



# Ecohydrological travel times derived from in situ stable water isotope measurements in trees during a semi–controlled pot experiment

David Mennekes[1,2], Michael Rinderer[1], Stefan Seeger[1], Natalie Orlowski[1]

[1] Hydrology, University of Freiburg, Freiburg im Breisgau, Germany

[2] now at: Empa − Swiss Federal Laboratories for Materials Science and Technology, Technology and Society Laboratory, St. Gallen, Switzerland

*Correspondence to*: david.mennekes@empa.ch

**Abstract.** Recent advances in in situ measurement techniques for stable water isotopes offer new opportunities to improve the understanding of tree water uptake processes and ecohydrological travel times. In our semi–controlled experiment with 20–

year–old trees of three different species (*Pinus pinea*, *Alnus incana* and *Quercus suber*) placed in large pots, we applied in situ probes for stable water isotope measurements to monitor the isotopic signatures of soil water and tree xylem before and after two deuterium labelled irrigations. Additional destructive sampling of soil and plant material complemented the in situ measurements and allowed for a comparison between destructive (cryogenic vacuum extraction and direct water vapour equilibration) and in situ isotope measurements. For the first labelling pulse, the tracer based travel time at a stem height of 15

cm was 0.7 days for all three tree species but at 150 cm height tracer based travel times ranged between 2.4 (for *Alnus incana*) and 3.3 days (for *Quercus suber*). The tracer based travel time from the root zone to 15 cm stem height was similar to the sap flow based travel times (i. e., for all trees 0.7 days). However, sap flow based travel times were 1.3 days (for *Alnus incana*) longer than tracer based travel times at 150 cm stem height. In terms of different between tree species, we found similar tracer movement in *Pinus pinea* and *Alnus incanca* while in *Quercus suber* tracer travel times were longer which is likely due to

lower water uptake rates of *Quercus suber*. The comparison of destructive and in situ isotope measurement techniques suggests notable differences in the sampled water pools. In situ measurements of soil and xylem water were much more consistent between the three tree pots (on average standard deviations were by 8.4 ‰ smaller for $\delta^2$H and by 1.6 ‰ for $\delta^{18}$O for the in situ measurements) but also among the measurements from the same tree pot in comparison to the destructive methods (on average standard deviations were by 7.8 ‰ and 1.6 ‰ smaller for $\delta^2$H and $\delta^{18}$O, respectively). Our study demonstrates the

potential of semi-controlled large scale pot experiments and high-frequent in situ isotope measurements for monitoring tree water uptake and ecohydrological travel times. It also shows that differences in sampling techniques or sensor types need to be considered, when comparing results of different studies and within one study using different methods.

## 1 Introduction

Rapid and small-scale processes driving interactions in the soil–plant–atmosphere system have only recently gained attention

and are, so far, neglected in most ecohydrological models. Challenges are partly the lack of sufficiently resolved data and





process understanding such as the temporal origin of precipitated water in the soil that is used by different tree species is e. g. in temperate ecosystems (Brinkmann et al., 2018; Sprenger et al., 2016a; Volkmann et al., 2016b). Thus, scaling down to a single tree or plant, ecohydrological processes related to water uptake and usage are not yet fully understood (Mahindawansha et al., 2018; Sprenger et al., 2019). A widely used tool in ecohydrological research are stable water isotope applications. They

allow to investigate and quantify ecohydrological processes such as plant water uptake depth and patterns and with that link plant water to its putative water source(s), such as groundwater, precipitation, stream water or (mobile or tightly bound) soil water (Goldsmith et al., 2012; Kübert et al., 2020; Orlowski et al., 2016b; Rothfuss and Javaux, 2017). This is possible because each water pool has its own unique stable water isotope signature due to underlying physical or chemical fractioning processes (Dubbert et al., 2019; Ehleringer and Dawson, 1992; Evaristo et al., 2015). It is widely accepted that stable water isotopes

($^2$H/$^1$H and $^{18}$O/$^{16}$O) are not altered by adsorption, degradation or delayed when water is taken up by roots (Ehleringer and Dawson, 1992). However, exceptions exist (e. g. Ellsworth and Williams, 2007; Zhao et al., 2016).

Recent studies working with stable water isotopes suggested highly complex plant water uptake mechanisms and patterns, which is in contrast to often used simplified soil plant water uptake models (e. g. Allen et al., 2019; Brooks et al., 2010; Evaristo et al., 2015; Lawrence et al., 2011; Sprenger et al., 2019; Wigmosta et al., 1994). In turn, a profound process understanding

and therefore more adequately constrained modelling will not only give crucial information on ecohydrological functioning but may subsequently serve as a robust virtual experimental platform. This allows for substantial improvements of the understanding of relevant e.g. forest ecosystem processes. However, quantitative databases combining continuous and spatially highly resolved samplings of all components relevant for the ecosystem water cycle are lacking.

Traditionally isotope information of soil water and xylem has been obtained by destructive sampling, lab–based water

extractions (e.g. for cryogenic vacuum extraction (hereafter abbreviated as cryo)) and consecutive isotope measurements of the extracted water. Destructive extraction methods are critically discussed in the literature in terms of accuracy and effects of soil parameters or organic contamination on the isotope measurements (Araguás-Araguás et al., 1995; Gaj et al., 2017a, 2017b; Millar et al., 2018; Orlowski et al., 2016a, 2016b, 2018a; Sprenger et al., 2015a; Walker et al., 1994). Generally, lab–based water extraction methods rely on destructive sampling of soil and plant material which is not capable for repeated

measurements at the same location and is strongly limited in temporal and spatial resolution of measurements (Kübert et al., 2020). However, it is not controversial that for fully understanding ecohydrological feedback processes and interactions, a more detailed temporal resolution over a longer observation period is necessary.

Such limitations can be overcome by in situ measuring methods for stable water isotopes which are more and more used in the ecohydrological community. An extensive review on in situ measurement methods for stable water isotopes can be found in

Beyer et al. (2020). So far, many in situ measurement methods are based on the vapour equilibrium principal (Wassenaar et al., 2008) and consist of gas permeable membranes (tubes or probes) through which water vapour is directed to a isotope analyser for real–time stable water isotope measurements (Gaj et al., 2016; Marshall et al., 2020; Oerter et al., 2016; Rothfuss et al., 2013; Volkmann et al., 2016a; Volkmann and Weiler, 2014). The sampled water vapour is assumed to be in equilibrium





with the liquid water surrounding the probes (soil or xylem). Water isotope standards (liquid and/or soil) or the equation by
Majoube (1971) are applied to transfer vapour to liquid stable water isotope ratios and to calibrate the obtained isotope data
(see Beyer et al., 2020). Membrane–based systems demonstrated not to cause isotope fractionation due to the passage of water
vapour through the membrane. However, one important point to be considered is that (depending on the type of isotope
analyser) a considerable amount of air is drawn from the soil or xylem media to be measured.

Continuous in situ stable water isotope measurements can provide data in high temporal resolution for relatively low
monitoring costs with high accuracy (Volkmann and Weiler, 2014). Thus, these methods potentially can provide the measuring
accuracy needed to clarify the ongoing discussions about "two water world"–like hypotheses (Berry et al., 2018) and help to
unravel high temporal and spatial dynamic processes occurring at the soil–plant–atmosphere interfaces. Such processes can
further encompass changes in plant water storage and water travel times from the tree to the ecosystem scale. However, in situ
measurements have been applied to trees in only very few cases so far (Marshall et al., 2020; Volkmann et al., 2016a).
Interpreting these results is sometimes challenging due to the complex boundary conditions that are given in natural forest
environments. An application on a range of species with different vessel and wood anatomies as well as applications over
longer time periods such as weeks or even months, is also still lacking.

Here, we use in situ stable water isotope probes developed and tested by Volkmann and Weiler (2014) and Volkmann et al.
(2016) in a semi–controlled outdoor pot experiment with 20 year old trees over the duration of 10 weeks. We test the application
of the new in situ measuring method for three different tree species having different anatomies (diffuse–porous vs. ring–porous,
shallow vs. deep root system). The aim of our research was to learn more about ecohydrological travel times, namely the time
water travels along the soil plant continuum from the roots to the tree stem and further to the canopy. So far, tree water uptake
is most commonly indirectly estimated by using sap flow velocities measured at one location of the stem (often breast height)
to derive the travel time that water travels from the roots to the canopy. We hypothesise that measuring the actual breakthrough
of a tracer (e.g., deuterium) that is transported in the tree xylem is a more direct measurement of ecohydrological travel times.
However, for detecting such an isotope tracer breakthrough, high frequency measurements at the same monitoring location
over multiple weeks are a prerequisite. This cannot be accomplished via destructive sampling of soils and tree xylem. The
tracer–based travel time approach has the advantage that it can be applied between any two points of the soil–root–stem–
branch water flow system of a tree. The temporal delay with which the labelled soil water appears in the tree stem and twigs
will allow for new insights into plant water uptake strategies. This delay has often been neglected when relating soil water
isotope composition to the isotope composition in trees to derive root water uptake profiles.

In specific, we tested the following research questions:

- Can the new in situ stable water isotope monitoring technique capture the isotope tracer arrival during a controlled
labelling experiment and thus allow to derive ecohydrological travel times in the soil–tree continuum?





- Can the new in situ stable water isotope monitoring technique be successfully applied to different tree species having different anatomies?

- How do the in situ stable water isotope measurements compare to destructive sampling techniques (cryogenic vacuum extraction and direct water vapour equilibration) that previously have been used for stable water isotope measurements in tree xylem and soils?

## 2 Methods

### Experimental setup

Our experimental set up consisted of a semi–controlled outdoor pot experiment with three approximately 20 year-old, 4 to 6 m high trees. One was a coniferous tree, i.e *Pinus pinea,* and two were deciduous trees i.e., *Alnus incana* and. *Quercus suber*, which from here on are referred to as *Pinus (P), Alnus (A)* and *Quercus (Q)*. The trees were planted into large pots (1.3 m x 0.75 m x 0.5 m) and the soil was covered with rain–out shelters (Fig. 1). The pots were filled with a fertile loess soil (Ut2, slightly clayey silt, according to the German Soil Classification KA5, Table 1) from a field site in Malterdingen, Germany (N 48.1592087, E 7.7823271). To monitor soil water conditions, we installed soil moisture sensors (Decagon 5TE sensors, Labcell Ltd, USA) in 15 and 30 cm soil depth and soil matric potential sensors (MPS sensors, Decagon Devices, USA) in 15 cm soil depth. In the same soil depths (15 and 30 cm) we also installed the in situ stable water isotope probes (for technical details and installation procedure see section about in situ probes) which in the following they are called S15 and S30. Stable water isotope probes were also installed in 15 and 150 cm stem height in trees' xylem (called X15 and X150 in the following). Trees were equipped with sap flow sensors (SF3 3–needle HPV Sensor, East 30 Sensors, USA) slightly above the isotope probes to avoid sensor measurement influences. All data were logged every 10 minutes. Meteorologic conditions (i.e., air temperature and precipitation) were measured on site (technical campus of the University of Freiburg, Germany) and at an official weather station of the German Weather Service (DWD) less than 1 km away (station id: 1443, N: 48.0232, E: 7.8343, elevation 237 m, temporal resolution: 10 min). A scaffolding was set up around the tree pots in order to be able to install and maintain the setup and take destructive samples during the experiment (Fig. 1).

**Table 1: Characteristics of the soil used for the experiment. Note that grain size distribution was analysed after removing the carbonate content with hydrochloric acid.**

| Parameter | | Value |
|---|---|---|
| | coarse sand | 0.5% |
| | middle sand | 1.9% |
| Grain size distribution | fine sand | 7.4% |
| | silt | 81.3% |
| | clay | 8.9% |





| German Soil Classification (KA5) | | Ut2 |
|---|---|---|
| Carbon content | fine grained | 18.5% |
| pH | | 8.2 |
| Max. water storage capacity* | | 63.5% |

120             * Gutachterausschuss Forstliche Analytik (2015)

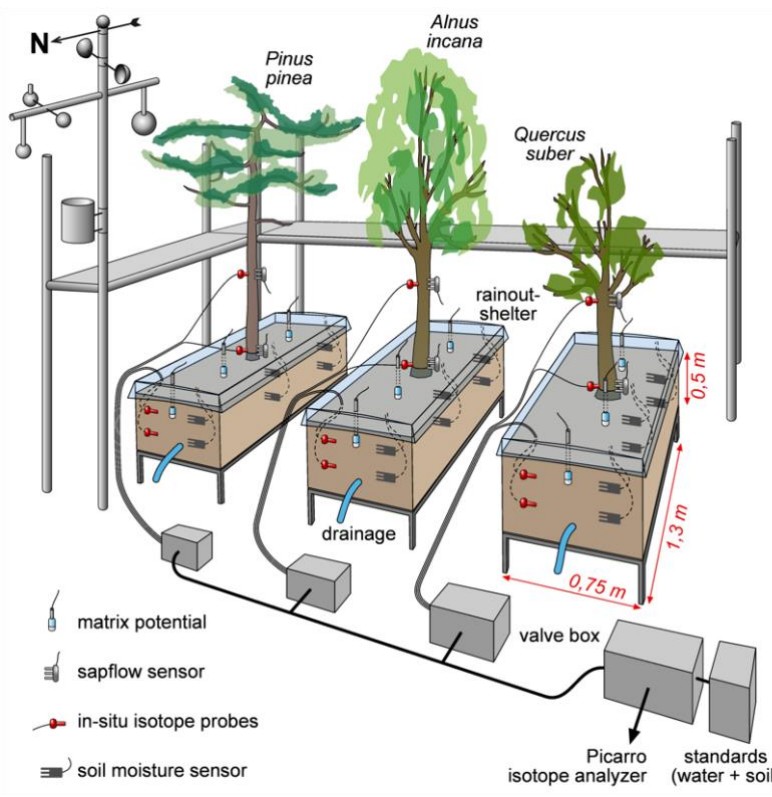

**Figure 1: Experiment set-up including used sensors and probes as well as the valve and tube set-up for the stable water isotope probes. Trees were exposed to outdoor conditions except for natural rainfall which was to prevent unknown water input into the soil. Note that scales for tree heights or equipment are not representative.**

**Irrigation labelling experiments**

The experiment consisted of two irrigation / labelling events in fall 2019 with two differently deuterated waters (+40 ‰ $\delta^2$H and +95 ‰ $\delta^2$H, Table 2). During the four weeks prior to the experiment trees were irrigated only once with water of a known isotopic composition to reduce the amount of antecedent water in the soil. The first labelling took place in the evening of 7 September and in the morning of 8 September 2019 with two times 20 mm labelling water (sum = 40 mm) with an isotopic

signature of +40 ‰ $\delta^2$H (Table 2). During the first labelling, we split the total amount of label water to be able to better monitor the arrival in soil water content. This procedure did not prove to be necessary and was thus not applied for the second labelling,





which took place in the evening of the 11 November 2019. We applied a total amount of 40 mm water with an isotope signature of +95 ‰ $\delta^2$H. For both labelling, we did not modify the $\delta^{18}$O composition (Table 2). To avoid saturated soil conditions at the bottom of the pots, a drainage system was installed but no drainage water was captured from these outlets during the entire

experiment.

**Table 2: Characteristics of the irrigation and labelling water during the experiment (note: The first labelling was split into labelling 1a and 1b to better monitor the response in soil moisture. During the second labelling this procedure was no longer necessary). \* manual event-based samples taken at a roof-top sampling station at the Chair of Hydrology laboratory, University of Freiburg, Germany. Amount refers to total precipitation.**

| Applied water | Date and time (local time) | Amount [mm] | $\delta^2$H [‰] | $\delta^{18}$O [‰] |
|---|---|---|---|---|
| Precipitation before experiment \* | 2 July 2019 to 20 August 2019 | 155 | -28.8 | -4.2 |
| Irrigation tap water | 4 September 2019 | 20 | -63.1 | -9.2 |
| Labelling 1a | 17 September 2019, 21:15 | 20 | +41.2 | -9.2 |
| Labelling 1b | 18 September, 10:30 | 20 | +41.2 | -9.2 |
| Labelling 2 | 10 October 2019, 20:30 | 40 | +95.1 | -9.2 |

**In situ stable water isotope monitoring**

For the in situ stable water isotope monitoring, we used probes similar to Volkmann and Weiler (2014) but with a shorter probe

head (length: 30 mm). The probes consist of two parts: a hydrophobic microporous polyethylene probing tube (outer diameter: 10 mm; length: 30 mm; Porex Technologies, Germany) and a mixing chamber (Volkmann et al., 2016a). A carrier gas (synthetic dry–air) was directed at a rate of 35 ml min⁻¹ through a Teflon tube into the tip of the porous membrane of the probe. There, the carrier gas equilibrates with the vapour in the xylem/soil surrounding the probe. Consequently, the equilibrated carrier gas was directed via Teflon tubes into a cavity–ringdown isotope analyser (Picarro L1102-*i*, Picarro Inc., USA). To

prevent condensation in the Teflon tube, the equilibrated carrier gas was diluted with synthetic dry–air in the mixing chamber to lower the vapour content of the sampling gas. The total flow rate was 35 ml min⁻¹ of which 10 ml min⁻¹ were diluted during the measurement. Before each in situ stable water isotope measurement, the tubing system was flushed for 10 to 30 min to prevent contamination of the isotope measurement by residual moisture in the tubing system. For more technical details the reader is referred to Volkmann and Weiler (2014).



For the probe installation in the tree xylem, the bark was removed and a 3 cm deep hole with a diameter of 1 cm was drilled into the tree stem. The holes in the trees were deep enough to fit the micro porous membrane heads of the isotope probes. The hole with the probe was finally sealed airtight with silicon, which was left to dry for a couple of days before the first isotope measurements.

To allow for later conversion of the vapour isotope measurements into liquid water isotope values, we used a set of isotope standards on site. The isotopic signature of these standards was repeatedly measured over the course of each day to be able to later compensate for any effect on isotope measurements that was potentially caused by applying the method in an outdoor and not in a lab environment. We used two types of standard boxes: Airtight PVC-tubes filled with soil and added water of a known isotopic signature (standard) and airtight PVC-tubes with an air filled headspace over the isotope standard in liquid form. For each soil standard 1200 g soil of the same soil as in the tree pots was dried at 200 °C for 24 h (cooled down in a desiccator) and filled into costume–made PVC- U boxes (KGU 125, Ostendorf KG, Germany; length: 14 cm, diameter: 12.5 cm) (Orlowski et al., 2013, 2016a, 2016b; Sprenger et al., 2016b Beyer et al., 2020). The soil was rewetted with 300 ml of three different standard waters (Table 3). The water standard tubes were filled with 150 ml of water from the same three standards. In each standard tube, one in situ isotope probe was installed and the tubes were sealed airtight. The standard tubes were placed in a Styrofoam box to exclude sun light and minimize temperature effects. Soil moisture content and temperature in each standard tube were monitored using 5TE probes (Labcell Ltd, USA). This setup enabled the compensation for any effect of daily temperature fluctuation on the isotope measurements as a matter of the outdoor setting of the experiment.

**Table 3: Soil and water standards for in situ stable water isotope measurements.**

| Name | Type | $\delta^2H$ [‰] | $\delta^{18}O$ [‰] |
|---|---|---|---|
| stdSL / stdWL | soil / water low | -82.0 | -11.4 |
| stdSM / stdWM | soil / water middle | -31.9 | -8.2 |
| stdSH / stdWH | soil / water high | +19.9 | -5.0 |

For reliable isotope measurements, the in situ isotope probe and the connected tubing were first flushed with synthetic dry–air for 10 to 30 min. Then a valve system facilitated a continuous flow of sampling gas from the in situ isotope probe to the Picarro isotope analyser. Measurements in the sampling mode were carried out until the isotope readings reached a stable plateau. A final data point was calculated based on the median of all reliable measurements of the last 4 min of stable readings. Reliable data were defined by criteria such as isotopic signal in possible range and successful flushing with dry–air.

The standard's vapour isotopic signals were related to liquid isotopic signals by measuring the liquid isotopic signal of the standards before and after the experiment (liquid ring–down spectrometer or cryogenic vacuum extraction and vapour equilibrium method). In a second step, the continuous measurements of the standards were used to transform measured in situ





stable isotope measurements from vapour to liquid isotopic signals. Isotope data were further corrected for changes in the humidity using the approach by Brand et al. (2009). Finally, the entire isotope dataset was further checked for outliers. We used an agglomerative clustering method combined with a single linkage method (Almeida et al., 2007; Hawkins, 1980) for

each isotope probe's data to identify potential outliers. Nearest neighbours were detected with Euclidean Distance and maximum clusters were numbers of observations divided by 10. Outliers were manually checked and removed if they were deemed to be unreliable.

**Destructive sampling: Cryogenic vacuum extraction and direct water vapour equilibration method**

One aim of our work was to compare the in situ stable water isotope measurements with isotope values obtained by commonly

used destructive water (vapour) extraction (equilibration) methods. We chose the cryogenic vacuum extraction (Araguás-Araguás et al., 1995) and the direct vapour equilibration methods (Sprenger et al., 2015a, 2015b; Stumpp and Hendry, 2012; Wassenaar et al., 2008), which have not been compared with such long and continuous in situ plant water isotope values before.

For obtaining isotope samples for the vapour equilibration method, twigs were collected from the tree crown at four different days during the experiment (Table 4). The bark was removed from the twigs and the samples (at least 30 g) were placed in

airtight coffee bags (1 L volume, Item no. CB400-420 BRZ, Weber Packaging, Germany). Soil samples were taken from soil cores from 0 to 25 cm and 25 to 50 cm soil depth using a Pürckhauer. Again, samples (at least 30 g) were placed in airtight bags. The sampling bags are comprised of tri–ply, adhesive laminated sheets including a 12 mm layer of aluminium foil and a zip–seal both guaranteeing air–tightness. Sprenger et al. (2015a) showed the suitability of these bags for soil samples and subsequent isotope analysis.

**Table 4: Dates of destructive sampling campaigns for consecutive cryogenic vacuum extraction (”cryo”) (Orlowski et al., 2018b) and the direct water vapour equilibration method (”direct equilibration”) (Wassenaar et al., 2008). Samples were collected before noon and directly stored in airtight bags/glass tubes and kept cool (4 °C) until further analysis.**

| Nr. | Date | Type | Method |
|-----|------|------|--------|
| 1 | 16 September 2019 | soil & xylem | direct equilibration & cryo |
| 2 | 02 October 2019 | soil & xylem | direct equilibration |
| 3 | 28 October 2019 | xylem | direct equilibration |
| 4 | 14 November 2019 | soil & xylem | direct equilibration & cryo |

The airtight sampling bags filled with the twig or soil samples were inflated with dry–air (pressure about 1 bar) and sealed.

The dry–air was then allowed to equilibrate with the moisture of the twig/soil sample for one (twigs) or two (soils) days at 20 °C lab temperature, respectively. For stable water isotope measurements of the air–tight bags, a silicon drop was placed on the bag, dried for a day and then punctured with a syringe that was attached to a lab-based isotope analyser (Picarro L2120-*i*, Picarro, USA) via a Teflon tube. To transfer vapour isotope values into liquid isotope values, three standards of 10 ml each



(FSM: $\delta^2$H = -126.2 ‰, $\delta^{18}$O = -16.7 ‰; tap water: $\delta^2$H = -65.9 ‰, $\delta^{18}$O = -9.6 ‰; Baltic Sea: $\delta^2$H = -2.6 ‰, $\delta^{18}$O = -0.4 ‰)

were filled in airtight bags and the isotopic composition was measured in the same way as for the twig / soil samples.

For cryogenic water extraction, we used a custom built set up at the laboratory of the Chair of Tree Physiology, University Freiburg (for further information see Dubbert et al., 2014, 2017). This facility participated in the round–robin by Orlowski et al. (2018b). The sampled twig and soil samples were placed into gas–tight 12 ml septum–capped glass vials (LABCO, United Kingdom) and heated for 90 min in a 95 ℃ hot water bath under a vacuum of at least 0.08 mbar. Extracted vapour was trapped

in glass tubes cooled with liquid $N_2$. After defrosting, samples were filtered.

Liquid isotope samples (from irrigation water, labelling water, soil and plant water extracts) were analysed using a L2130-i laser spectroscope (Picarro Inc., Santa Clara, CA, United States) in the laboratory of the Chair of Hydrology. All runs included three in–house standards, which were calibrated against V-SMOW, SLAP, and GISP (IAEA, Vienna) following Newman et al. (2009). This further allowed for drift and memory corrections. Isotopic ratios are reported in per mil (‰) relative to the

Vienna Standard Mean Ocean Water (VSMOW) (Craig, 1961). The deuterium excess was calculated as $d = \delta^2$H $– 8 * \delta^{18}$O (Dansgaard, 1964).Precision of analyses was ±0.6 ‰ for $\delta^2$H and ±0.16 ‰ for $\delta^{18}$O. Isotope data were checked for spectral interferences using ChemCorrectTM (Picarro Inc., Santa Clara, CA, United States). In our study, no plant or soil water sample was found to be affected by organic contamination.

For the fourth measurement campaign (Table 4), stable water isotope signatures of cryogenically extracted water were

additionally measured with a mass spectrometer (Delta plus XP; Thermo Finnigan, USA) following the measurement routine described by Saurer et al. (2016) (precision: 1.5 ‰ for $\delta^2$H and 0.2 ‰ for $\delta^{18}$O).

**Sap flow data post–processing**

Heat pulse signals of the sap flow sensors (SF3 3-needle HPV Sensor, East 30 Sensors, USA) were converted into velocities following the Eq. (1) by Hassler et al. (2018) who derived their equation from (Campbell et al., 1991):

$$V_{sap} = \frac{2k}{C_w \ (r_u + r_d)} \ln\left(\frac{\Delta T_u}{\Delta T_d}\right) \tag{1}$$


where $k$ is the thermal conductivity of the sapwood, set to 0.5 W m$^{-1}$ K$^{-1}$, $C_w$ is the specific heat capacity of water in J m$^{-3}$ K$^{-1}$, $r$ is the distance between heating (middle) and measuring (outer) needles in m (here 6mm) and $\Delta T$ is the temperature difference before heating and 60 s after the heat pulse in K. Indices $d$ and $u$ stand for downwards and upwards compared to the heated needle in the middle.

Sap flow velocity was additionally corrected for wounding of the xylem tissue and installation effects using Eq. (2) according to Burgess et al. (2001):



$$V_c = b\,V_{sap} + c\,V_{sap}^2 + d\,V_{sap}^3 \tag{2}$$

where $V_c$ (m s$^{-1}$) is the corrected $V_{sap}$ and $b$, $c$ and $d$ are the correction coefficients. In line with Hassler et al. (2018), we set $b$ = 1.8558, $c$ = −0.0018 s m$^{-1}$ and $d$ = 0.0003 s$^2$ m$^{-2}$ (Burgess et al., 2001; Hassler et al., 2018). Outliers were removed by applying a 30 min rolling median to the dataset. General offsets were corrected by shifting values in a way that nightly sap
flow velocities were zero following Pfautsch et al. (2010).

**Travel time estimation**

To analyse the isotope tracer arrivals after the labelling events, the tracer arrival times at the in situ isotope probes in 15 cm and 150 cm stem height were derived. The tracer arrival time was determined as the first time step after the labelling for which $\delta^2$H isotope values exceeded the range of isotope values in the 48h before the labelling. We also determined the time step of
the highest isotope measurement after the labelling (called peak time in the following). Due to a short data gap right after the second labelling in November we did not include these data but focussed our analysis only on the first labelling experiment.

We compared the tracer based travel times with sap flow based travel times for which we calculated cumulated sap flow using sap flow velocity data of *Pinus* and *Alnus* sap flow sensors in 15 cm and 150 cm stem height. The sap flow data of *Quercus* was considered as not reliable and excluded from the analysis. We speculate this could be due to the ring porous sap wood
structure (see also Bush et al., 2010; Clearwater et al., 1999). We then determined the time step for which cumulated sap flow was equal to the travel distance from an average rooting depth of 15 cm below surface to 15 cm and 150 cm stem height (i.e. 30 cm and 165 cm travel distance).

## 3 Results

**Dynamics of climatic, soil moisture and sap flow conditions**

During the experiment, average air temperature at the experimental site was 14.0 ℃ (sd: 4.4 ℃) ranging from 4.0 ℃ on the coldest day to 28.2 ℃ on the hottest day (Fig. 2). Daily maxima were recorded at noon ranging from 8.2 ℃ to 28.2 ℃ (mean: 18.5 ℃, sd: 3.2 ℃), while minima were measured around 4:00 in the morning. Vapour pressure deficit (VPD) as relevant plant activity indication was highest between 13:00 and 15:00 with average values around 8.5 kPa (sd: 0.6 kPa), reaching maximum values around of 2 kPa for a few days. Soil temperatures in both depths of all pots followed air temperatures trends, although
the amplitude was approximately ten times less pronounced and up to 10 h delayed. During the experiment, the mean soil temperature for all sensors was 15.1 ℃ (sd: 2.7 ℃).





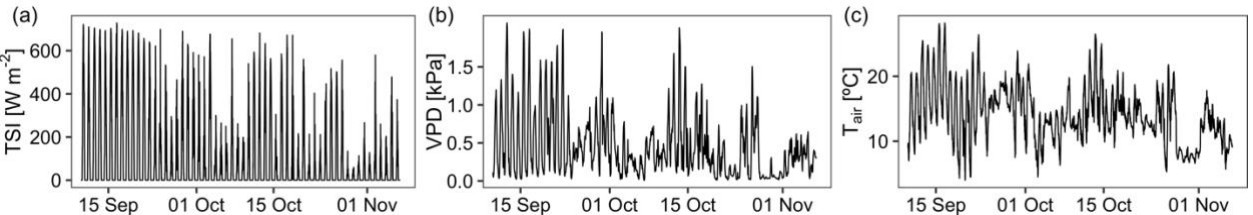

**Figure 2: Weather conditions for the study site over the course of the experiment. Data is provided by the German Weather Service station Freiburg i. Br., Germany, less than 1 km away.**

Soil moisture dynamics were only affected by artificial irrigation and the labelling campaigns, since rain–out shelters prevented natural precipitation to enter the tree pots. The installed drainage system at the bottom never showed any outflow. The irrigation with labelled water caused the average volumetric water content (VWC) to increase by 24 % across all trees (Fig. 4). Soil moisture differed between sensors of the same pot and was highest for *Quercus* (range of average VWC values: 12.2 % – 32.4 %), and smaller for the *Pinus* pot (9.6 % – 24.5 %) and the *Alnus* pot (13.0 % – 18.5 %) (Fig. 4). Over the course of the

experiment, median VWC in each pot was highest for *Quercus* (20.8 %) and lower for *Pinus* (11.7 %) and *Alnus* (14.2 %). This suggests that root water uptake of the *Quercus* tree was lower than for the other trees. This is also reflected in the matric potential values that exceeded permanent wilting point (PWP) more often for *Alnus* and *Pinus* than for the *Quercus* soil pot (Fig. 4). In 77 % of all soil moisture profiles across all tree pots we observed higher VWC at 15 cm than at 30 cm soil depth.

Soil matric potential showed a similar dynamic compared to VWC: starting at pF values around 2 after labelling/irrigation and

increasing during dry periods (up to 3.2 for *Pinus*, 4.2 for *Alnus* and 3.2 for *Quercus* before the second labelling; up to 4.4 for *Pinus*, up to 5 (measurement limit) for *Alnus*, and 3.5 for *Quercus* at the end of the experiment). Generally, pF values increased fastest and strongest for *Alnus*, while matric potentials in the *Quercus* pot showed the smallest increase (max pF value 3.4). Over the experimental period, median pF value was 2.4 for *Pinus*, 3.2 for *Alnus* and 2.1 for *Quercus*,, respectively (Fig. 4). The highest sap flow velocities were measured in *Pinus* at 150 cm height (max 12 cm h$^{-1}$, median = 1.9 cm h$^{-1}$), while *Alnus*

showed lower maximum sap flow velocities (at 150cm height: 7.4 cm h$^{-1}$, median = 0.8 cm h$^{-1}$) over the experimental period. A maximum sap flow velocity of 2.1 cm·h$^{-1}$ was measured for *Quercus*, which is likely due to sensor failure (Fig. 4). Therefore, we excluded this data from further analysis.

**Isotopic tracer arrival in soils and trees**

The $\delta^2$H signatures measured with all in situ isotope probes before the first labelling in soils and trees was ca. -30 ‰ and

therefore similar to the $\delta^2$H of summer precipitation of the preceding weeks (-29 ‰) (Fig. 3). One exception was the sensor *Pinus* X15 in 15 cm stem height which showed $\delta^2$H values of ca. -15 ‰. Here, it should be noted that the soil was exposed to the summer precipitation before the start of the experiment. The first labelling on the 17 September 2019 caused a rapid increase of the isotopic signature measured with the soil probes in each pot except for *Pinus* S30 and *Alnus* S15. A likely explanation is, that the labelling water was not penetrating the entire soil profile as homogeneously as intended. The *Alnus* S15





measurements showed a rather steady increase in $\delta^2$H values. About two weeks after the first labelling, *Alnus* S15 isotope values were equal to all other *Alnus* soil and xylem in situ measurements. In comparison, *Alnus* S30 isotope values responded within 15 h. The peak of the soil $\delta^2$H isotopic signature of S30 was +9.14 ‰ and therefore elevated by +45.15 ‰ compared to the signature from the day before the labelling. For *Pinus* S15 and S30, the response pattern was the opposite. S15 responded within 5 h and reached a peak of +8.77 ‰ for $\delta^2$H within a day (delta = +39.94 ‰ for $\delta^2$H) while *Pinus* S30 responded within

ca. 2 days and had only a peak of -19.08 ‰ for $\delta^2$H (ca. 2 weeks after the first labelling). *Quercus* S15 and S30 in situ soil probes responded both within half a day and both reached a peak within 1 day after labelling. This difference in the response of isotopic measurements in different soil depth suggests that in the *Alnus* pot our labelling water might have reached lower soil depth by preferential flow paths bypassing the soil matrix of the upper soil profile. In the *Pinus* pot the labelling water first infiltrated into the upper part and reached deeper soil depth with a considerable delay. In case of the *Quercus*, our irrigation

led to a homogeneous distribution of the label water across soil depth within half a day as intended for all pots in our experiment.

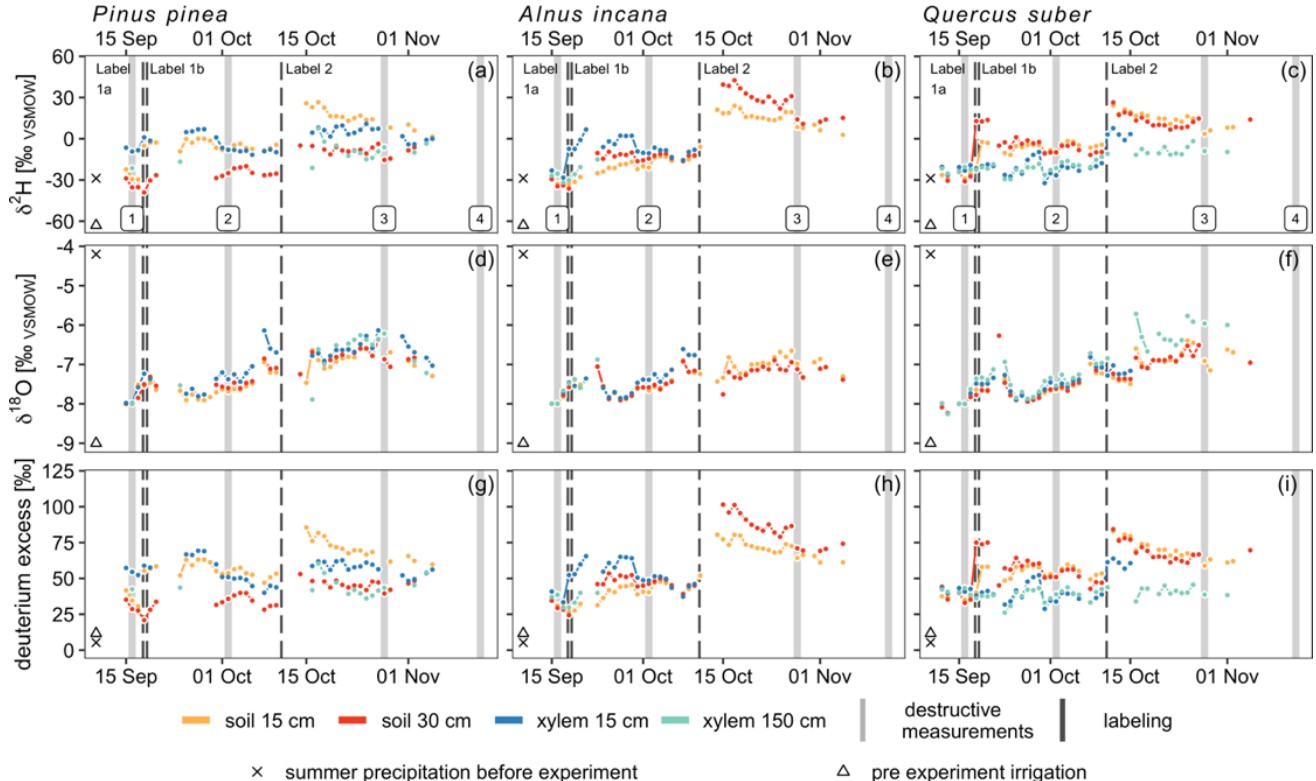

**Figure 3: Daily median stable water isotope measurements with in situ probes in xylem and soils. $^2$H and $^{18}$O are shown in δ notation while deuterium excess is calculated as follows: d = $\delta^2$H − 8 ∗ $\delta^{18}$O (Dansgaard, 1964). For reference, plots include destructive**
**measurement campaigns (light grey) and labelling dates (dark grey) as well as summer precipitation (cross) and pre event irrigation isotope signal (triangle).**





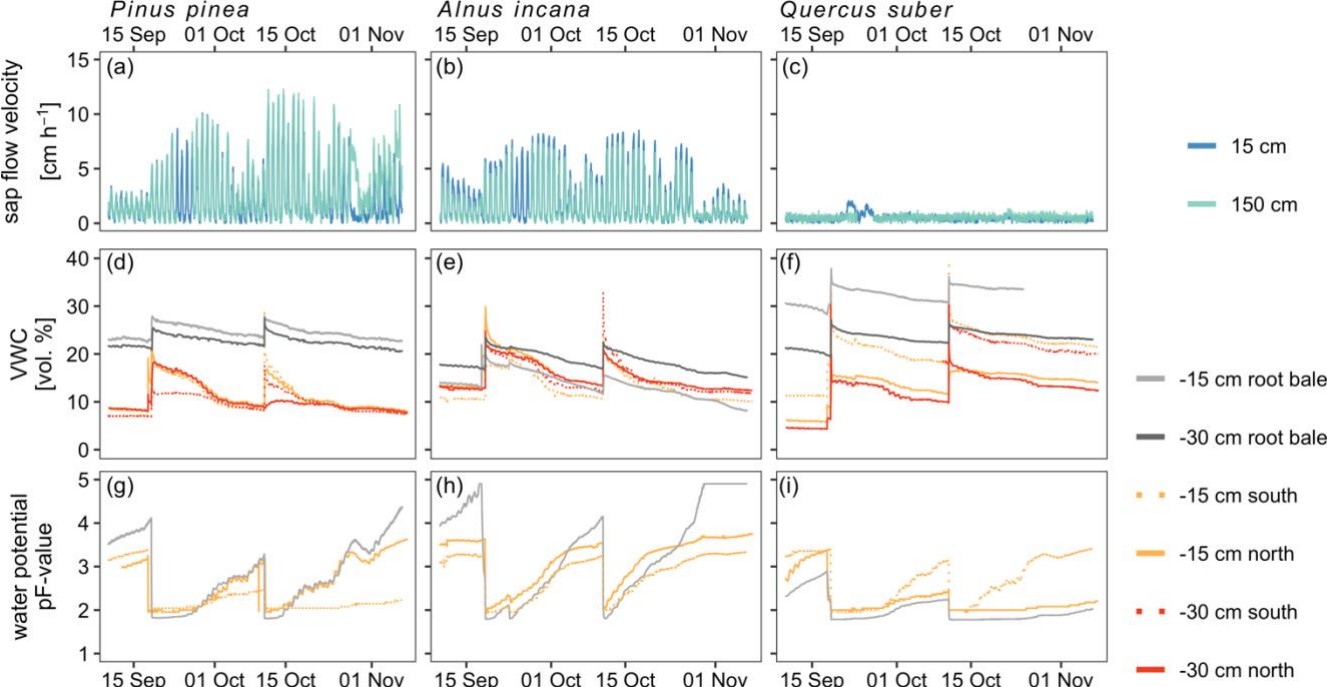

**Figure 4: Soil-, tree- and atmospheric conditions over the course of the experiment in 10 min temporal resolution. Soil condition and sap flow velocity is shown for the three tree pots. Sap flow velocity measurements for *Quercus suber* are most likely affected by**
**sensor failures (panel c).**

In situ isotope probes in tree xylem of *Pinus* X15 responded within 17 h and reached a peak 2.9 days after the first labelling (-15.11‰ for $\delta^2$H). *Pinus* X150 could not be evaluated because of a data gap shortly after the labelling. *Alnus* X15 also responded within 17 h and reached a peak ca. 3 days after the first labelling (+6.74 ‰ for $\delta^2$H) *Alnus* X150 responded within 2.4 days

and reached a peak after 5.9 days (-13.47 ‰ for $\delta^2$H). The response of isotope measurements in *Quercus* X15 and X150 was different from the two other trees. We could not identify a clear first arrival but instead the isotopic values increased slowly over time. Based on our definition of "first arrival" to be the time step the isotope values exceeded the range of isotope measurements of the 48 h prior to the labelling, the first arrival for *Quercus* X15 and X150 would have been after 17 h and the 2.3 days but we do not consider these "first arrivals" to be very reliable. More clear were the peaks for the $\delta^2$H isotope

measurements of *Quercus* X15 and X150, which we measured after 2.3 days (peak: -14.67 ‰ $\delta^2$H) for X15 and after 16 days (peak: -14.68 ‰ $\delta^2$H) for X150.

The travel time based on cumulated sap flow was very similar to our travel times based on first arrival in isotope measurements (Table 5). For instance, for *Alnus* X15 the isotope based travel time and the sap flow based travel time were both 17 h (Fig. 5). For *Alnus* X150, the isotope based travel time was 2.4 days, the sap flow based travel time was 3.7 days. Similarly, for *Pinus*

X15 the isotope based travel time and the sap flow based travel time were 17 h. We lack data to determine the isotope based


travel time for *Pinus* X150 but the sap flow based travel time was 3.5 days. For *Quercus*, we lack reliable sap flow data that is why we could not compare isotope based and sap flow based travel times. We discuss technical issues in Sec. 4 that led to data gaps but the experimental setup in general was found to be suited for measuring tracer breakthrough and ecohydrological travel times in the soil-plant continuum.

**Table 5: Tracer arrival and peak times in xylem isotope probes at 15 cm and 150 cm height in *Pinus*, *Alnus* and *Quercus* of the first labelling on 17 September 2019. In addition, an estimated time delay based on cumulated sap flow velocity data is given. It is calculated as the time needed for sap to travel from -15 cm average rooting depth to 15 cm and 150 cm stem height, respectively. NA: no isotope data available, X: no sap flow data available**

| Tree | Type | Height [cm] | $\delta^2H$ before labelling [‰] | $\delta^2H$ first arrival [‰] | difference between $\delta^2H$ before labelling and first arrival [‰] | Time delay first arrival [days] | Time delay based on sap flow [days] | $\delta^2H$ peak [‰] | difference between $\delta^2H$ before labelling and peak [‰] | Time delay peak [days] |
|------|------|------|------|------|------|------|------|------|------|------|
| *Pinus* | Soil | 30 | -36.40 | -30.42 | 5.98 | 1.9 | | -19.08 | 17.32 | 16.8 |
| *Pinus* | Soil | 15 | -31.17 | -11.80 | 19.37 | 0.2 | | 8.77 | 39.94 | 0.7 |
| *Pinus* | Xylem | 15 | -13.37 | 4.99 | 18.36 | 0.7 | 0.7 | 15.11 | 28.48 | 2.9 |
| *Pinus* | Xylem | 150 | -21.44 | NA | NA | NA | 3.5 | NA | NA | NA |
| *Alnus* | Soil | 30 | -36.01 | -31.20 | 4.81 | 0.6 | | 9.14 | 45.15 | 1.0 |
| *Alnus* | Soil | 15 | -35.88 | -30.50 | 5.38 | 0.7 | | -11.12 | 24.76 | 16.4 |
| *Alnus* | Xylem | 15 | -28.28 | -7.50 | 20.78 | 0.7 | 0.7 | 6.74 | 35.02 | 3.2 |
| *Alnus* | Xylem | 150 | -32.74 | -19.80 | 12.94 | 2.4 | 3.7 | -13.47 | 19.27 | 5.9 |
| *Quercus* | Soil | 30 | -32.74 | 3.26 | 36.00 | 0.1 | | 22.73 | 55.47 | 1.0 |
| *Quercus* | Soil | 15 | -30.39 | -21.22 | 9.17 | 0.6 | | 1.58 | 31.97 | 1.0 |
| *Quercus* | Xylem | 15 | -23.85 | -19.66 | 4.19 | 0.7 | X | -14.67 | 9.18 | 2.3 |
| *Quercus* | Xylem | 150 | -22.44 | -18.68 | 3.76 | 3.3 | X | -14.68 | 7.76 | 16.3 |

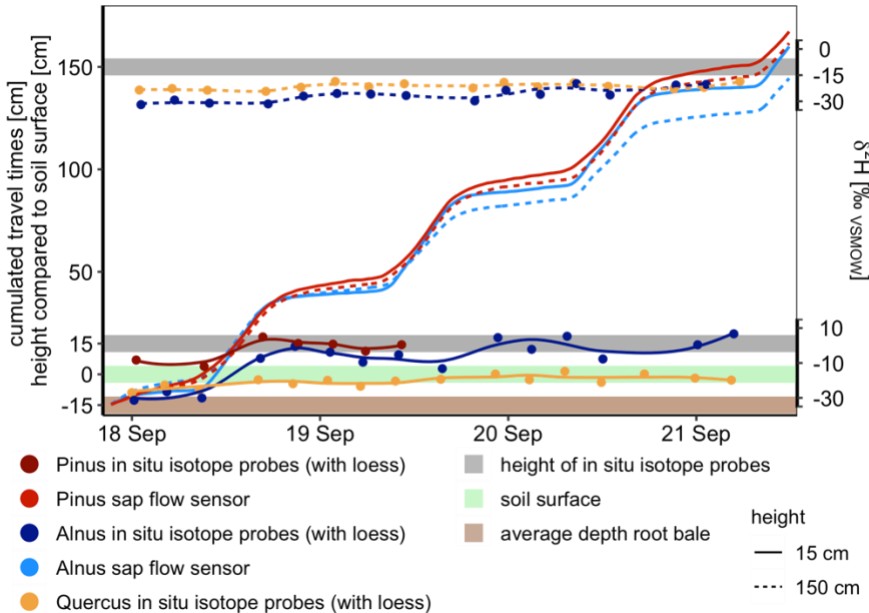

**Figure 5: Isotope tracer breakthrough of *Alnus*, *Pinus* and *Quercus* at 15 cm and 150 cm stem height and cumulated sap flow of *Alnus* and *Pinus* (no reliable sap flow data for *Quercus*). Tracer based travel times (i.e. time of first response of isotope tracers) for *Alnus* and *Pinus* were similar to sap flow based travel times considering the distance from an average rooting depth of 15 cm below surface to 15 cm and 150 cm stem height.**

**Comparison of in situ stable water isotope measurements and destructive sampling techniques**

When comparing in situ isotope data with destructive measurements, the later showed a wider spread within isotope measurements from the same tree species (soil vs. xylem/twigs) than the in situ measurements (Fig. 6). Hence, for one tree species (soil and xylem/twigs values) results for destructive methods showed maximum differences for $\delta^2H$ up to 49.2 ‰ during one measurement campaign, while $\delta^{18}O$ showed differences of up to 4.79 ‰.

Isotope values from in situ measurements of soils and xylem were much more consistent between the three tree pots (soil and xylem) but also among the measurements from the same tree pot, especially under natural abundance (measurements before labelling). This was particularly true for $\delta^{18}O$. For $\delta^{18}O$ under natural abundance conditions and low VWC, the differences between the three isotope methods were much more apparent and became less strong after the first and second labelling (smallest methodological differences). Xylem $\delta^{18}O$ values showed the smallest variation between the three methods for the 350 very last measurement campaign, even so VWC was comparable to the initial values at the start of the experiment (before the first labelling). The soil $\delta^{18}O$ values under natural abundance (before labelling) obtained from cryogenic extraction and the direct water vapour equilibration were much more enriched than in situ methods' $\delta^{18}O$ values. Average values across all measurements were -2.3 ‰ and -3.1 ‰ for direct water vapour equilibration and cryogenic vacuum extraction and -7.8 ‰ for the in situ measurements. Additionally, we observed a trend over time for $\delta^{18}O$ values. While for in situ measurements $\delta^{18}O$





values increased over time (from average -7.8 ‰ for measurement 1 to -6.9‰ for measurement 4; Fig. 3), δ¹⁸O values of
        destructive measurements decreased (from average -2.6 ‰ for measurement 1 to -6.2 ‰ for measurement 4, Fig. 3). For
        example, δ¹⁸O values for *Pinus* decreased on average over all probes (xylem and soil) by −2.69 ‰ between the first and last
        measurement campaign. For *Alnus*, this decrease was −3.77 ‰ and for *Quercus* −4.26 ‰ for δ¹⁸O. As a result, differences in
        δ¹⁸O values between measurement methods (in situ vs. destructive) became smaller than ±2.5 ‰ (for measurement campaign
three and four) (Fig. 6). Regarding δ¹⁸O, destructively sampled values suggested a greater influence of more enriched pre
        experiment precipitation (Fig. 3 and Fig. 6). This was only true for δ¹⁸O but not for δ²H.

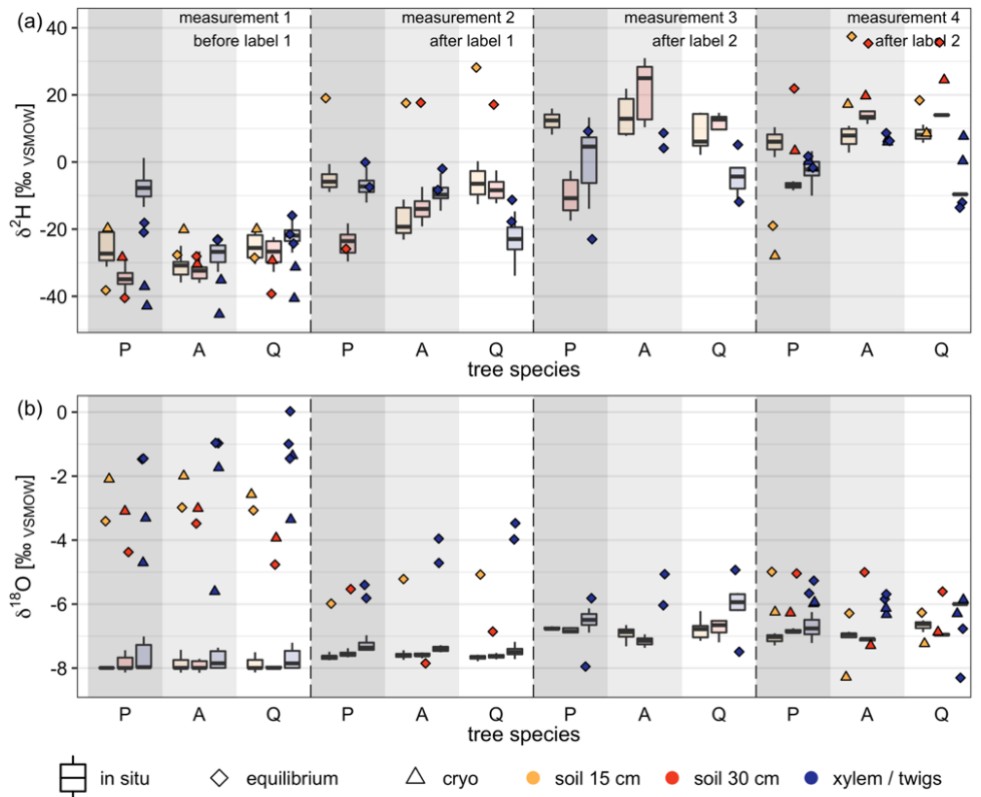

**Figure 6: Comparison of destructive isotope measurement methods (Direct water vapor equilibration = equilibrium, Cryogenic vacuum extraction = cryo) with in situ stable water isotope measurements. Box plots (with median) represent in situ measurements**
**for a six day (on average 11 measurements, range 1 to 41 measurements) time frame around the destructive measurement campaigns. Dates of measurement campaigns and applied methods can be found in Table 4.**
**Tree species at x-axis refer to: P = *Pinus pinea*, A = *Alnus incana*, Q = *Quercus suber*.**
**For reference, stable water isotope signature of labelling events, pre experiment precipitation and pre experiment irrigation can be found in Table 2.**






For $\delta^2$H values, 63 % of the destructive measurements lay outside the 95% confidence interval of the in situ stable water isotope measurements when considering a six days' timeslot (centralized on the day of destructive measurement). On average, δ-values for both isotopes were more positive for destructive measurements than for in situ measurements (Fig. 6).

Comparing the differences between destructive and in situ measurements for $\delta^2$H, we found significant differences between the four different measurement campaigns (ANOVA, p< 0.05). However, comparing differences between methods grouped by tree species or measured material (xylem or soil), no significant differences were found. Additionally, both destructive methods, cryogenic vacuum extraction and direct water vapour equilibration, showed different results for $\delta^2$H values than for $\delta^{18}$O. Furthermore, during the measurement campaign before labelling, soil xylem $\delta^{18}$O and $\delta^2$H values for cryogenic vacuum extraction were slightly more enriched than direct vapour equilibration values, while in xylem the cryogenic vacuum extraction values were mostly more enriched. On average, the measured values for soil samples for $\delta^2$H and $\delta^{18}$O were 13.23 ‰ and 1.35 ‰ more enriched for cryogenic vacuum extraction than for vapour equilibration. However, for xylem/twig measurements most of the δ-values obtained by the vapour equilibration method were higher than the values obtained by cryogenic vacuum extraction (for $\delta^2$H: 5 out of 6; for $\delta^{18}$O: 4 out of 6). Here, average absolute differences were 11.95 ‰ for $\delta^2$H and 1.56 ‰ for $\delta^{18}$O. Additionally, during the last measurement campaign for most samples the vapour equilibrium method showed more enriched $\delta^2$H and $\delta^{18}$O values.

In summary, we observed partly different results for $\delta^2$H and $\delta^{18}$O when comparing destructive with in situ isotope measurement plus an additional variability in destructive measurement observations. Furthermore, it should be noted that in situ observations contain multiple data points to avoid influence by daily fluctuation.

We cross-checked the cryogenically obtained isotope data on a mass spectrometer. Overall measured samples (N=12 for after the second labelling campaign), mean differences between the laser-based isotope measurements and the mass spectrometer measurements were 2.9‰ ±1.1 ‰ for $\delta^2$H and 0.0‰ ±0.3 ‰ for $\delta^{18}$O, which lies in an acceptable range of measurement inaccuracy. Thus, effects of potentially co-extracted organics can be ruled out.

## 4 Discussion

### In situ stable isotope measurement technique for monitoring ecohydrological travel times

One aim of our study was to evaluate the capability of the new in situ stable water isotope measuring technique (Volkmann and Weiler, 2014) in terms of deriving ecohydrological travel times. These ecohydrological travel times are particularly relevant to better understand water uptake strategies, source water depths and delays between root water uptake and evapotranspiration at the tree crown. We need more and better information on ecohydrological travel times to evaluate and further improve ecohydrological models (Knighton et al., 2020).





Our data suggest that the in situ isotope probes are capable to measure tracer arrival at different locations along the soil-plant continuum. In general, soil in situ isotope probes responded quickly to our applied isotope label suggesting that the technique is capable to detect changes in isotope composition in soils (see also Volkmann et al., 2016b; Volkmann and Weiler, 2014). In fact, they were capable to detect differences in isotope composition in different soil depth (e.g., *Pinus* and *Alnus* soil probes). The tracer arrival was clearly visible in the xylem isotope data of *Pinus* and *Alnus* at 15cm and *Alnus* at 150cm stem height

(data gap for *Pinus* X150, no immediate response in *Quercus* X15 and X150). This allowed to monitor the temporal dynamics of isotope signatures (i.e. tracer breakthrough) including a peak isotope signature at all probes in all three trees. In terms of this tracer breakthrough, we observed a flatter curve at the X150 probes than at the X15 probes suggesting that along the flow path diffusion occurred. This effect is also reported by others (Barbeta et al., 2020; Pfautsch et al., 2015; Schepper et al., 2012) and is likely to increase with increasing flow path length. For our experiment, the use of a tracer signal that was elevated by

ca. 70 ‰ compared to the isotope signature in soils and trees before the labelling (ca. -30 ‰) was enough to detect the tracer arrival in 150 cm stem height. Marshall et al. (2020) suggest to use an even more elevated tracer signal for a similar tracer experiment under controlled conditions and with mature trees.

Isotope measurements along extended flow paths might also be subject to potential isotope fractioning processes in the root zone, during water uptake or in the plant (Ellsworth and Williams, 2007; von Freyberg et al., 2020; Poca et al., 2019; Vargas

et al., 2017) which would further complicate the interpretation of tracer breakthrough signals. Others have discussed the influence of an exchange of water stored in wood cells and transported in tree xylem (Martín-Gómez et al., 2016). Further research is therefore needed to quantify the importance of these processes for isotope measurements in trees. However, for the detection of the first arrival and a tracer breakthrough of an isotope label, we argue that these processes are likely of minor importance.

**Isotope based versus sap flow based ecohydrological travel times**

In terms of ecohydrological travel times, the first arrival in isotope signature in 15 cm stem height of *Pinus* and *Alnus* could be detected 17 h after the labelling. The same duration was determined based on cumulated sap flow data. This result suggests that high–frequency in situ isotope measurements can be used to derive ecohydrological travel times in the soil-plant continuum. However, the difference in isotope based and sap flow based ecohydrological travel times increased with increasing

stem height. In 150cm stem height of the *Alnus* tree, the isotope based travel time was 2.4 days while the sap flow based travel time was 3.7 days. Thus, the question remains, which of the two travel times is more reliable. We acknowledge that the tracer signal in the tree xylem is a breakthrough curve that allows for some uncertainty in terms of how the first tracer arrival is defined. However, we argue that our definition of the first tracer arrival, namely a sudden increase in isotope signature that significantly exceeds the natural variability and measuring accuracy observed in the days prior to the experiments, is a reliable

indication of the tracer arrival. We argue that the isotope based travel time approach is a measure that integrates changing flow conditions along the flow path, e. g. short temporal water storages, while the sap flow based travel time is an extrapolation based on a point specific measurement. Our data suggest that this difference is trivial for short travel distances but can become





relevant already for measurements at breast height (150 cm). We expect these differences to become even more prominent when calculating travel times from the root to the canopy for entire forest stands (see also Meinzer et al., 2006;
Schwendenmann et al., 2010). It is likely that the isotope based and the sap flow based ecohydrological travel times differ by the order of several days or even weeks when comparing the two measures in 25 or 50 m tree height. Meinzer et al. (2006) compared sap flow base and deuterium tracer based measurements in coniferous species (*Pseudotsuga meziesii* (Mirb.) Franco and *Tsuga heterophylla* (Raf.) Sarg) with heights ranging from 13.5 m to 58 m and diameters between 0.14 m to 1.43 m. They found sap flow velocities to be five times smaller than tracer based sap flow velocities. Similar results were reported by
Schwendenmann et al. (2010) who studied topical trees and bamboo (4.2 m to 19.8 m height, 0.10 to 0.18 m diameter) and found sap flow velocities measured with a heat dissipation method seven times smaller than tracer based estimates. Parts of these differences can likely be explained by the fact that only a fraction of the sapwood cross-section consists of conduits and that the specific hydraulic conductivity of latewood is about one order of magnitude lower than that of earlywood (Meinzer et al., 2006) In other words, the sap flow based velocities are likely underestimating true sap flow. We argue that tracer based
ecohydrological travel times represent a more direct measure of the water transport in trees over longer distances than sap flow based travel times. The difference in isotope arrival between *Quercus* and the two other tree species in our study also suggests that the in situ isotope technique is capable to measure tree species specific differences in ecohydrological travel times.

**Differences between tree species**

So far, the in situ measurement method by Volkmann and Weiler (2014) was applied in soils (Volkmann and Weiler, 2014)
and a limited number of tree species (maple and European beech) (Seeger et al., 2020; Volkmann et al., 2016a). However, water uptake and transport varies for different tree species depending on multiple factors such as abiotic factors (Fonti and García-González, 2008), vessel width (Hagen-Poiseuille equation) and vessel structure. Thus, clear differences in conifers' xylem structure (e. g. for *Pinus pinea)* and hardwoods' xylem structure (e.g. *Alnus incana*, *Quercus suber*) exist. Conifers are dominated by tracheids which mainly transport water equally within the xylem cross section (Hacke, 2015). Hardwoods' xylem
consist of twisting non parallel traches (vessels) for nutrition and water transport and wood fibre for stabilization purpose (Kadereit et al., 2014). Here, vessel distribution and thus water transport within the xylem differs between (semi)-ring porous (e.g. *Quercus suber*) and diffuse porous (e.g. *Alnus incana*) tree species. While ring porous trees tend to build vessels in different sizes, vessels of diffuse porous trees are more equal in size, which also means a more equal water transport velocity distribution within the xylem (Barij et al., 2011; Kadereit et al., 2014; Leal et al., 2007).

We could identify differences in the dynamics of isotope tracer arrival between tree species that can likely be explained by differences in tree physiology (Fig. 4, Fig. 7 (a) and (b)). When differences of X15 minus the label/irrigation signal are calculated for both stable water isotopes and plotted in dual isotope space, negative $\delta^2H$ values suggest that the tree did not use much of the isotopically enriched water of the label (Fig. 7 (a) and (b)). Consequently, the trees take up a mixture of labelled and of non-enriched water pre experimental soil water (Table 2). In contrast, for X150, $\delta^2H$ values for all trees were
similar. This contrasts to the observed water uptake rates or sap flow velocities for the different tree species (Fig. 4, Table5).


However, for X15, the slower tracer arrival for *Quercus* (Fig. 3) is visible when looking at the $\delta^2H$ values, which are more negative compared to the other trees (Fig. 7 (a)). Overall, this suggests that with increasing height (X150) the isotopic signal of the water is more similar across all trees regardless of their water uptake quantities.

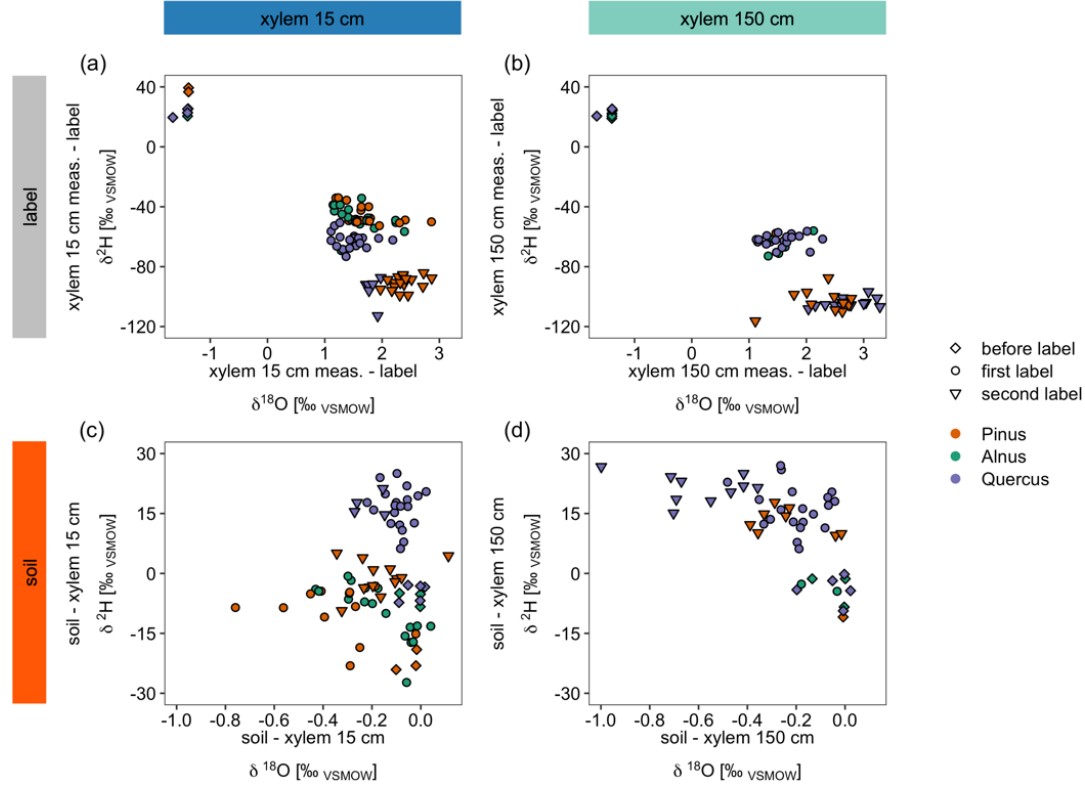

**Figure 7: (a) and (b) show differences between daily median in situ stable water isotope measurements and the label water's stable water isotope signature presented in a dual isotope plot. (c) and (d) compare daily median in situ soil measurements (averaged over both soil depths) with the corresponding xylem measurements of the same day.**
**Note that the points "before label" refer to the average value between pre event precipitation and pre event irrigation water while first and second label refer to the isotopic signal of the label water. See Table 2 for the isotopic compositions.**

The same differences can be calculated for the soils (averaged over both depths) and X15 or X150 (Fig. 7 (c) and (d)). Here, negative numbers in the dual isotope plots show isotopic enrichment in xylem compared to the soil. This is visible especially for $\delta^{18}O$. Differences for $\delta^2H$ in Fig. 7 (c) show that *Quercus* did not take up as much labelled water as the other trees, since the soil water is more enriched in $^2H$ than the xylem water. This is consistent with our previous findings. However, differences between before the first labelling, after the first labelling and after the second labelling measurements were less obvious than in Fig. 7 (a). Fig. 4 suggests that soil water at S15 and S30 was not fully replaced by the labelling water. In case of complete water replacement Fig. 7 (c) would be similar to Fig. 7 (a). Furthermore, Fig. 7 (d) suggests similar to Fig. 7 (b) and Fig. 5 that the tracer arrival in X150 was less pronounced than in X15 since soil water was comparably more enriched in $^2H$ than xylem



water. The decrease of tracer visibility in X150 was stronger for *Pinus* than for *Quercus* (for *Alnus* not sufficient data were observed).


We recommend to manipulate only one isotope ratio of water (either hydrogen or oxygen) since this holds the chance for a backup and controlling with the not manipulated isotope (here $^{18}$O).

We argue that information about vessel structure should be considered for analysing in situ isotope measurements. Furthermore, tree reaction, such as blocking traches due to wounding should be considered, especially if this causes less xylem

water flow with stable water isotope tracer passing by the in situ stable water isotope probes installed in the tree xylem. Similar observations for different species, e. g. installation of sap flow sensors or impact of outer forces, are shown in literature (e. g. Ballesteros et al., 2010; Barrett et al., 1995; Burgess et al., 2001; Schmitt and Liese, 1993). This is generally consistent with our observations which show less sensor failure for sap flow sensors in the coniferous tree compared to the semi-ring porous *Quercus suber* (Fig. 4, Fig. 6).

For further research, we suggest that in situ stable water isotope probes should be tested in a range of different tree species to better understand influences of tree properties such as xylem structure, reaction to wounding, soil conditions etc. However, more repetitions per tree species are also indispensable to improve statistical accuracy, since even trees from the same species may respond differently. Using the shorter probe heads (3 cm) in our study compared to Volkmann and Weiler (2014) (5 cm) ensured the measurement in the sapwood with active sap flow, which normally takes place in the outer 3 cm of the tree trunk

(Caylor and Dragoni, 2009; Cohen et al., 2008; Hacke, 2015).

**Comparison of in situ and destructive isotope measurements**

The second aim of this work was to systematically compare the in situ isotope measurements with isotope values gained by destructive sampling. In our study the destructive measurements showed a wider spread of isotope measurements in soil and xylem than the in situ isotope measurements (Fig. 6). Particularly before and during the first labelling experiment the $\delta^{18}$O

isotope values differed considerably between the in situ method on the one hand and the cryogenic vacuum extraction and the equilibration method on the other hand. This is likely due to sampling water from different water pools in soil and xylem. The effect of the sampling method on the isotopic signature of a sample is heavily discussed within the ecohydrological research community (Berry et al., 2018; Beyer et al., 2020; Bowers et al., 2020; Kübert et al., 2020; Orlowski et al., 2019; Penna et al., 2018; Sprenger et al., 2018). In our study this effect is clearly visible for $\delta^{18}$O and less for $\delta^{2}$H. Cryogenic vacuum extraction

is known to sample bulk soil water but also hygroscopic and biologically bound water (Koeniger et al., 2011; Orlowski et al., 2016b; Sprenger et al., 2015a). On average, $\delta$–values for both isotopes were more positive for the destructive measurements than for in situ measurements in our study. In a method comparison by Orlowski et al. (2016b) over different soil types, the direct water vapour equilibration method showed higher $\delta^{18}$O and $\delta^{2}$H values in comparison to cryogenic vacuum extraction.





Previous studies have further shown that the direct water vapour equilibration method can show a wider spread in isotope data
across the same soil type (Orlowski et al., 2019), especially in low water content soils (<5 % gravimetric water content of less
than 3 g of water in the sample) and consolidated shales (Hendry et al., 2015; Wassenaar et al., 2008). However, in our study
with only few data points for the destructive methods, we did not find a general trend of higher $\delta^{18}O$ and $\delta^2H$ values for the
direct water vapour equilibration method.

In our study $\delta^{18}O$ measurements of the destructive sampling during the first measurement campaign suggest the samples
contained more tightly bound water that was similar to pre experiment precipitation (Table 2, Fig. 6). This was most apparent
for $\delta^{18}O$ but not for $\delta^2H$. The applied $^2H$ label in our study likely has masked the effect of sampling from different water pools
for the $\delta^2H$ measurements. Kübert et al. (2020) found small differences between in situ measurements and cryogenically
extracted isotope values under natural abundance conditions. However, after a strongly enriched $^2H$ label application, the
authors observed considerable differences between cryogenically extracted and in situ soil water vapour measurements
(following the method of (Rothfuss et al., 2013)). In our study, the observed differences in the $\delta^{18}O$ values measured with the
in situ isotope probes and destructive sampling methods became smaller over time (Fig. 6). This is most likely due to the fact
that the water used for irrigation/labelling (e. g. $\delta^{18}O$ = -9.2 ‰) became more abundant in smaller pores over the course of the
experiment and the amount of pre experiment precipitation ($\delta^{18}O$ = -4.2 ‰) became smaller over time. This assumption is
supported by the more negative $\delta^{18}O$ values of the destructive samples towards the end of the experiment (Fig. 6). For the last
measurement campaign, all isotope methods showed the least differences in $\delta^{18}O$.

In a lab comparison of extraction systems for water isotope analysis from spring wheat, Millar et al. (2018) also found
considerable differences between the applied methods. The tested methods (direct vapour equilibration, microwave extraction,
two versions of cryogenic extraction systems, centrifugation, and high-pressure mechanical squeezing) showed considerable
differences depending on the extracted plant part (head, stem, leaf, root crown) and the isotope ($^2H$, $^{18}O$). The direct vapour
equilibration method showed the smallest standard deviations for leave material (±1.48 ‰) and the highest for head plant
material (±6.92 ‰) for $\delta^2H$. For $\delta^{18}O$, root crown material produced smallest standard deviations (±0.64 ‰) and head plant
material largest (±3.41 ‰), for laser–based isotope analysis on an IWA–45EP analyser (Los Gatos Research Inc., Mountain
View, CA, USA) for the same method. A comparable cryogenic extraction system to the one used in our study, produced
highest standard deviations for leaf material (±2.81 ‰) and smallest for root crown material (±0.93 ‰) for $\delta^2H$. For $\delta^{18}O$,
smallest standard deviations were measured for head material (±0.31 ‰) and largest for leaf material (±1.02 ‰). These results
highlight the influence of methodological challenges, even when applying the same method to samples from identical plant
species (i.e. spring wheat). Such differences may also vary depending on the soil moisture conditions (shown in a study by
Kübert et al. (2020) under perennial grassland) and therewith transpiration rates and sap flow velocities of the individual plants.
In our study, $\delta^2H$ differences between the two xylem measurements per measurement campaign averaged to 3.4 ‰ (*Alnus*),
8.3‰ (*Quercus*) and 11.4 ‰ (*Pinus*) for the vapour equilibrium method. Average differences for the cryogenic vacuum
extraction were in a similar range (*Pinus* = 3.8 ‰, *Alnus* = 5.6 ‰, *Quercus* = 8.3 ‰, Fig. 6). For $\delta^{18}O$, we found a smaller



variability for the vapour equilibrium method measurements (*Pinus* = 0.47 ‰, *Pinus* = 0.74 ‰, *Quercus* = 1.52 ‰) compared to cryogenic vacuum extraction (*Pinus* = 0.72 ‰, *Alnus* = 2.03 ‰, *Quercus* = 1.21 ‰, Fig. 6).

VWC conditions during the first and last measurement campaign were similar. Thus, a potential effect of VWC would be similar for all methods but has been shown to strongly influence lab-based methods results (Orlowski et al., 2016b, 2019).

Millar et al., (2018) further argue that cryogenic vacuum extraction accesses the total plant water, resulting in more depleted $^2$H and $^{18}$O. For the first measurement campaign in our study, $\delta^{18}$O values from cryogenic vacuum extraction were also depleted in comparison to both equilibrium methods (Fig. 6) for the xylem water of all three tree species but more enriched for the soil water samples. For the last measurement campaign, xylem water $\delta^{18}$O values plotted much closer together (with smallest differences between the methods) and only the *Pinus* and *Alnus* trees showed more negative values. For $\delta^2$H and the first measurement campaign, xylem water was also more depleted overall tree species, but these differences were almost negligible for the last measurement campaign. $\delta^2$H values of the *Quercus* tree obtained by cryogenic vacuum extraction were even more positive in comparison to the other methods (consistent with the $\delta^{18}$O values).

Since our in situ measurements showed overall the smallest differences over varying soil moisture and sap flow conditions and is applicable in both soils and trees, we recommend using in situ real-time measurement methods for future ecohydrological studies. Only with such methods, highly dynamic processes at the soil-tree interface can be monitored with high frequencies over long time periods and no methodological artefacts are produced when different methods are used for the soil and tree compartment. Nevertheless, if high–frequency sampling is not the first priority, the direct vapour equilibration method is favoured over cryogenic vacuum extractions for known methodological issues produced by the cryogenic vacuum extraction method, in particular for soils. In addition, the direct vapour equilibration method is comparably cheaper and easier to use than the cryogenic vacuum extraction method (Kübert et al., 2020; Orlowski et al., 2016b).

**Experimental setup**

Our semi-controlled outdoor pot experiment aimed to minimize potential influences of boundary conditions that typically occur in natural forest environments e.g., soil heterogeneity, subsurface flow and redistribution of soil water, rainfall input, stemflow and associated variability in isotope signatures in the soil–tree compartment (see von Freyberg et al. (2020). Typically, variability in outdoor conditions are overcome by working in a greenhouse under controlled conditions with e.g., tree seedlings instead of adult trees and a homogenous substrate. However, tree seedlings considerably differ in their physiognomic properties compared to mature trees. This was the reason why we used 20 year old 4 to 6 m high trees that are more similar to mature trees.

In general, the variability in our isotope data has shown that outdoor isotope measurements are challenging and that the quality is not comparable to the accuracy gained with lab–based isotope measurements of liquid water. The use of on–site standard boxes (water and soils) that were exposed to the same environmental conditions improved the accuracy of our isotope measurements. The high temporal frequency with which one can measure the isotopic compositions with in situ isotope probes



allows for quantification of this environmental variability and for efficient averaging to gain reliable results. We argue that the accuracy is well suited to perform tracer labelling experiments and monitor the tracer arrival that is typically orders of magnitude larger than natural isotopic variability. Studies that used destructive sampling also reported variability of isotope measurements taken from one single tree (von Freyberg et al., 2020). This is due to the fact that destructive sampling relies on multiple sampling locations that can differ (naturally) in their isotopic signature. Using stable water isotope tracers is motivated by the possibility to better understand plant water uptake and usage. Most often, ecohydrological studies rely on water

extraction methods for isotope analysis with low temporal and spatial resolution, e. g. cryogenic vacuum extraction, which can also be limited in precision (see Orlowski et al., 2016b). Our in situ measurement setup was able to measure the stable water isotope composition in xylem and soil water in high-resolution (> 60 per day) over months and relate this to further monitored environmental parameters (e.g. matric potential, VWC, sap flow). High temporal resolution constitutes an important step towards better understanding of fractioning processes or mixing of different water pools in soils and trees (von Freyberg et al.,

2020; McDonnell, 2014).

**Future implications**

Long–term high resolution isotope data are essential for advancement in future ecohydrological investigations. They will provide a broader understanding of water fluxes in the soil–plant–atmosphere continuum especially when temporal resolution of isotope measurements will be fine enough to resolve potential intraday variability. This could further allow the observation

of temporal storage changes, dispersion processes (in soils and potentially trees) and differences in water flow path ways through individual plants and related water age differences (Sprenger et al., 2019); although this is still highly discussed in literature (e. g. Berry et al., 2017; Dubbert et al., 2019; Sprenger et al., 2016a, 2018).

The in situ stable water isotope measurement method presented here guarantees stable measurements of isotope values in high temporal resolution over months. As the same method can be applied in soils and tree xylem at the same time, consistency and

methodological comparability is given in terms of sampling of the same water pools (Beyer et al., 2020; Volkmann et al., 2016a). However, in-situ outdoor isotope measurements in general are still challenging and often do not reach the accuracy of lab–based measurements of liquid water samples. Further development of the experimental equipment is needed to better compensate for these environmental effects (mainly temperature variation) and to increase the measurement accuracy. Our method is suitable for tracer labelling experiments but also works well under natural abundance conditions since those were

the times where the smallest methodological differences compared to destructive methods were observed. In future research, the applied in situ method will be tested on a wider range of tree species and be tested for its applicability in other vegetation types (e.g. shrubs, vegetables, fruits). For now, more replicates would have been desirable and required to better estimate the measurement accuracy and reliability under different climatic conditions and soil water availabilities. However, studies like ours and work by von Freyberg et al. (2020) contribute to the discussion of potential changes in the isotope signature during

plant water uptake and water flow within the plant xylem related to fractionation processes.



## 5 Conclusions

We present a semi–controlled outdoor isotope labelling experiment with three different tree species (*Pinus pinea, Alnus incana*, and *Quercus suber*). Our aim was to test the applicability of a new in situ measurement approach in soils and tree xylem and to compare the measurements to two destructive sampling techniques (cryogenic vacuum extraction and direct water vapour equilibration). The obtained isotope data were further used to derive ecohydrological travel times in the soil–tree continuum and to compare those to travel times derived from sap flow measurements. Our high–frequency isotope measurements captured the arrival of the added isotope tracer in different soil depth and stem heights of our 4 to 6 m high trees. Our measurements of $^2$H and $^{18}$O compositions in soils and trees, allowed to identify tree species specific differences in ecohydrological travel times from the roots to 15 cm and 150 cm stem height. The tracer based travel times for shorter travel distances (i.e. 15 cm stem height) were 0.7 days over all three tree species and also similar to travel times derived from sap flow data. However, the difference between the two travel time approaches became larger with increasing travel distance (i.e. 150 cm stem height). For instance, *Alnus incana* tracer based travel time was 1.3 days shorter than sap flow based travel times. We therefore expect travel times based on sap flow measurements to be potentially different by several days when estimating water travel times from the roots to the canopy of a natural forest which might reach up to 40 m in height or more. Thus, we consider the isotope tracer–based travel time to be a more direct measure of the actual water flow towards the canopy. Our isotope method comparison showed that isotope values from in situ measurements of soil and xylem water were much more consistent between the three tree pots (for both soil and xylem) but also among the measurements from the same tree pot in comparison to the destructive methods (cryogenic vacuum extraction and direct water vapour equilibration), especially under natural abundance conditions. Thus, standard deviations for destructive measurements were on average by a factor of 1.7 higher for $\delta^2$H and 4.5 for $\delta^{18}$O than in situ measurements when comparing measurements between the three tree pots. For the same tree pot (soil and xylem water), on average standard deviations were by a factor of 1.6 higher for $\delta^2$H and 4.9 for $\delta^{18}$O for destructive measurements in comparison to in situ measurements. Hence, methodological differences were much more apparent for the "non–labelled" $\delta^{18}$O values. The differences between the three isotope methods became less strong after the first and second deuterated water labelling. We assume that the cryogenic extraction method extracted soil water that was dominated by the isotope signature of the precipitation before the experiment while the in situ isotope values reflected the signature of the less tightly bound water pools that had less time to interact with the soil matrix (i.e. the labelling water). Our results are comparable to other studies that have shown the effect of methodological differences on sampled soil and plant water pools and their associated isotope signatures. Such methodological differences currently complicate the interpretation and comparison of isotope results from different studies (varying in e.g. tree species, soil moisture conditions, applied methods). Our in situ isotope method is however capable of measuring the isotopic signature of soil and tree xylem water in high temporal resolution from different tree species and soil depths at the same location over several month. This is a prerequisite for advancing our understanding in terms of plant source water depth and source water age and mixing of different water pools along the flow path from the roots to the canopy.



## Data availability statement

The datasets generated for this study are available on request to the corresponding author.

## Competing interests

The authors declare that they have no conflict of interest.

## Author contributions

Funding acquisition: NO, MR, SS; experimental design: NO, MR, SS; field measurements: DM, SS, NO, MR; Data analyses: DM, MR; manuscript preparation: DM, NO, MR, SS. All authors reviewed the manuscript.

## Funding

This study was funded by "Freiburg's Academic Society". The article processing charge was funded by the Baden-Wuerttemberg Ministry of Science, Research and Art and the University of Freiburg in the funding programme Open Access Publishing.

## Acknowledgments

This work was funded by "Freiburg's Academic Society". We thank Barbara Herbstritt for lab support and student intern Bernhard Gigler for his help during field campaigns. Hugo de Boer is thanked for plant physiological advice and his help during the experiment setup and Britta Kattenstroth for technical support. We further thank the Chair of Ecosystem Physiology for being able to use their greenhouse and cryogenic vacuum extraction facility, the Chair of Hydrology for technical field equipment and Jun. Prof. Dr. Matthias Beyer for the provision of the *Alnus* and *Quercus* trees.



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
