# Peer review of "Ecohydrological travel times derived from in situ stable water isotope measurements in trees during a semi-controlled pot experiment"

_Hydrology and Earth System Sciences, 2020_

## Author Response (AR1)

Reviewer comments in Black, Responses in Blue.

**Reviewer 1**

We thank Referee #1 for the comments, which were very useful to prepare an improved version of the manuscript. We answer below to each comment in a point-to-point reply.

**General comments**

**1)**
Thank you for this comment. We agree and connected the first part of the introduction better with the importance of the here presented in situ method. Furthermore, we mention and transfer arguments in the introduction again in the discussion.

**2)**
Thank you for this comment. We shortened the results, e.g. for the climate, soil conditions subsection we deleted more specific air temperature information.
For the isotope label arrival section, we summarized / shortened information about specific arrival patterns (starting L292).
We changed the order of results in the subsection "comparison of in situ vs destructive", talking first about 2H and then 18O which is more consistent with the other text.
Regarding figure 6: We understand the problematic of the figure. We tried to rearrange axes and categories. However, we think that in the current version one can see best, especially for 18O, that differences between methods (in situ vs. destructive) are much higher than between species. Furthermore, one sees well the variability for 2H. Regarding differences between the cryogenic extraction and water vapor equilibrium method, we agree that differences are hard to detect in the graph. However, we argue that this was not our main focus since here also more data would be needed for a fundamental research.

**3)**
Thank you for this comment. We generally shortened the discussion and start with an introductory sentence now. Regarding your specific points:
**Avoid repeating the aims/goals of the study. This information should be in the Introduction**
We did delete the repetition of our aims/goals.

**Start the Discussion with an opening paragraph in which in a couple of sentences, you report the two or three most relevant results of the study but without going into much detail. Importantly, compare these main results with the most relevant literature. Currently this would be in the paragraph starting at L400, but I am not sure that you are putting everything relevant there. Obviously, my proposal for this first paragraph is questionable. In any case, I find it useful for the readers to make it easy to find the most relevant results and conclusions in the beginning of the Discussion.**
Thank you for your suggestions. We followed your suggestion and start the discussion with an introductory sentence.

**Try to avoid as much as possible the repetition of the results (L475-484 but check the rest).**
We checked the discussion for possible repetition and deleted those where possible.

**Do not report others' results if they are not clearly linked to yours (Millar et al. 2018).**
Thank you for your remark. We deleted this section since it is clearly not relevant for our paper.

**Structure the paragraphs (not only the subsections) with clear and identifiable concepts, do not leave isolated two-line sentences.**
We improved the structure of this paragraph.

**The Conclusions are well-written but somewhat repetitious with the rest of the Discussion. I would use this briefer conceptual scheme for the rest of the Discussion, while embedding the relevant literature. In any case, I think that the Discussion should be shortened, retaining only the most relevant points to discuss.**
Thank you for this comment. We have combined conclusions and implications into one section and considerably shortened it

**Specific comments**
Abstract (L9): Perhaps I miss a sentence on the main goal of the experiment.
Rephrased the beginning of the abstract (L9 – 11) to make our main goal clearer.

L21.
Thank you for pointing this out. It is based on the measured stable water isotope signals. We added this to the sentences.

L30-32
We rephrased the sentences to: "Challenges are partly the lack of sufficiently resolved data but also the lack of ecohydrological process understanding, e. g. the origin of water used by different tree species (Brinkmann et al., 2018; Ellsworth et al., 2007; Sprenger et al., 2016a; Volkmann et al., 2016b)."

L35
Thank you for your comment. We rephrased the sentences according to your suggestion.

L37
We agree with your comment and rephrased the sentences: "For the separation of water pools based on the concept that potentially each water pool has its own unique stable water isotope signature due to underlying physical or chemical fractionation processes, highly precise and / or frequent stable water isotope measurements are needed (Dubbert et al., 2019; Ehleringer and Dawson, 1992; Evaristo et al., 2015)."

L40-41
We totally agree with your comment and changed the text.

L42
We deleted the study by Wigmosta et. al. in our citation.

L42-48
We added some words to the paragraph to make our message more precise.

L49
Thank you for your suggestion! We changed information to "composition"

L50
Yes, "consecutive" was the wrong word. We changed it to "subsequent".

L52
Yes. We changed it according to your suggestion.

L56

We changed the beginning of the sentences to: "However, it is undebatable that…."

L66-68
we rephrased the sentences to: "Furthermore, for membrane–based systems no isotope fractionation could be observed when water vapour passes through the membrane. However, it should be considered that a considerable amount of air / vapour is withdrawn from the soil or xylem media by the necessary flow rates of the isotope analyser."

L93-94
Thanks. We added the word "water"

L127-L128
Yes maybe… However, we also wanted to start the experiment a bit earlier but ran into some troubles just before starting. Next time we would use a soil with better known isotopic composition. On the other hand, since the "old" water wasn´t completely replaced we found some interesting results regarding the plant water uptake.

L130
We changed the sentences to: "During the first labelling, we split the total applied amount of label water into two rounds of irrigation to be able to better monitor the arrival in soil water content"
We hope that the sentences is clear now.

212
We added the relevant information.

L280
Partly yes. A college of us, Simon Haberstroh, works a lot with cork oak and sap flow sensors. He told us that likely the tree would inactivate damaged sapwood tissue which somehow would be a sensor failure. He observed partly similar behaviour for his trees. However, we didn´t really find a paper which did publish data to sensor failure. Unfortunately, talking about non-working experiments is not very popular… We are also aware that the result fits partly with the general inactivity of the oak tree. However, we also measured photosynthesis and vapour conductance (not mentioned in the paper) and both values were more or less normal. Summarized, we did not really consider the cork oak for our results but instead of excluding the tree completely one can learn about possible difficulties and potential improvements in future..

Figure 3
Yes, it would be helpful. However, the second label is far above the limit of the x-axis which would cause quite some white space. We did add the number as text.

L341-344
we rephrased the sentences to: "when comparing in situ isotope data with destructive measurements, the later showed a wider spread of isotope measurements within"

L378
the sentence was removed…

L374-385
the paragraph was rephrased.

L388-392

Thank you!

L396
We followed your general comments and rephrased the beginning of the discussion.

L403
Yes, thank you for mentioning. We changed it to depths

L408
we added the sentence: "This could be caused by temporal trapped water or different flow path lengths and velocities or potential isotopic fractioning effects reduced the isotopic tracer signal"

L416
Yes. We added Barbeta et al. 2020 here.

L457
Thank you for your important comment. We are aware of the related problems of using a Mediterranean species in Germany and added the sentence "it should be mentioned that *Quercus suber* is a Mediterranean species and does not naturally grow in Germany."

L478
We added: ", e. g. slower decrease in VWC after irrigation (Fig. 4)" to support the statement

L480-484
This paragraph was rephrased

L486
This paragraph was rephrased

L502-506
This paragraph was rephrased

L509
This paragraph was rephrased

L511
Thank you for your question. We did not test for significant differences here because of the limited number of datapoints. We rephrased the sentence to make this clear.: For instance, we found, in general, that δ–values for both isotopes were mostly more positive for the destructive measurements than for in situ measurements.

L531-543
This paragraph was deleted.

L549-550

We joined the sentences to another paragraph.

L551-568
We did connect the results by Millar with our study by adding the following sentences:
"However, in our experiment we found that most δ-values of cryogenic vacuum extraction were more negative than from the water vapour equilibration method (Fig. 6). Nevertheless, we are aware of our small destructive sampling size and therefore focus less on differences between both destructive measurement methods."

L536-656
Sorry, we don´t understand your comment. Also, the line numbers might be wrong / misleading.

L568
Thank you for your valid comment and your argumentation to move this section into the method section. However, we would like to keep this section in the discussion part to get higher reader attention regarding future possible experimental set-ups. Consequently, we here want to discuss and present our thoughts why we did our experiment the way we did it.

**Reviewer: 2**

We thank referee #2 for the valuable comments, which helped us to improve the manuscript. Please find the answers to each comment below.

**General comments**

We generally shortened the discussion and added more logical connections to our results. Overall we also improved the understanding to find some clear take home messages, especially regarding the results section.

**Specific comments**

L39-41
Thank you for your comment. We changed this part to better highlight the recently discovered fractionation processes.
Furthermore, in our discussion we talk about possible effects of fractionation occurring in our experiment when we discuss Fig. 7 in section "Differences between tree species".

Figure 5
Thank you for your suggestion. We agree that the way we chose the y2-axis makes it difficult to see the dynamic of the isotope data (because dynamics is damped).. To improve this, we changed (i.e., stretched) the y2-axis to better show the dynamics of the isotope data. However, we think that plotting isotope data and cumulated sap flow data into the same figure allows a better comparison of the timing of isotope break through and the sap flow based ecohydrological travel time (i.e., the time it took sap to travel from the roots to 15 cm and 150 cm tree hight, respectively). This is what we try to emphasize with the brown, green and gray horizontal bar.

L480,

Indeed, this was confusing. We rephrased this section and we think it should be clearer now.

L485-486
We change the paragraph regarding Fig. 7. We hope it will be clearer now.

L485-486
As mentioned above, we changed the discussion part. We hope it is understandable now.

L549-550
We removed the two-line paragraphs.

Conclusion
Thank you for your idee. We appreciated it and transformed the conclusion to a section called "conclusion and future implications". Consequently, we also hope that we avoid too many redundant information and everything becomes shorter and more precise.

**Technical comments**

L165
Thanks a lot for pointing out this typo! We changed it to custom-made

L267
Thank you for this technical comment. We are aware of this.
In the revised version, we would like to have the two graphs on one page if possible. In this case we wanted to have the more interesting result on top. If the final edited version won´t be like this, we will change the order.

L481-484
We rephrased this section and we think it should be clearer now. See comments above and comments, too.

**References**

---

## Author Response (AR2)

Reviewer comments in **black**, responses in **blue**

**Reviewer 2:**

We thank Referee #2 for the comments which will help to prepare an improved version for publication. We answer below to each comment in a point-to-point reply.

Specific comments:
- L46-54: this paragraph seems convoluted and I am not sure it add much to the introduction, since the need the high-frequency measurements has already been stated. I would suggest to remove this paragraph, or condense it in one sentence.

Thank you for this comment. We agree that the information is redundant and we did delete the paragraph.

- L86: "The aim of our study is to..."?

We changed „was" to „is"

- L453-454: The sentence, describing deuterium depletion, seems to run counter to proposed argument so as to why there could be deuterium enrichment in the xylem. Further, I cold not find Zhao et al. (2017) but if the correct reference is Zhao et al. (2017) (https://doi.org/10.1111/pce.12753), then this paper does not discuss an ubiquitous response of xylem deuterium signature (as compared to the phloem, depending on the xylem's location, or compared to soil signature). As a result, I would suggest to simply remove this sentence which does not really support the discussion one way or another.

We agree that the study by Zhao et al. is a bit misleading in the discussion. We replaced the sentences by a more general sentence to provide a more general discussion on possible fractioning / enrichment effects on stable water isotopes: "Another source of isotope enrichment in xylem water could be fractioning processes during or after plant water uptake. Thus, data from before the labelling experiment show for all trees isotopic enrichment in the xylem water compared to the soil water isotopic composition (Fig. 7 (c) and (d))."

According to the Reference Zhao et al., 2017: It should be Zhao et al., 2016.

- L498-501: At first glance, it seems to me that the findings in Kübert al. (2020) are opposite to the one reported here, since here the authors first report more obvious d18O differences between in situ and destructive sampling, and d18O is precisely the isotope under "natural abundance". One can however tease out the "considerable differences" between methods found for soil d2H after label1, but it seems that direct equilibrium values are more enriched, running counter to the hypothesis of having more depleted pre-event water sampled as compared to in situ method ?

Please rephrase these sentences to better capture the point of convergence/divergence between that study and yours.

We do not fully understand the reviewer's comment. We have changed the sentence to: "Kübert et al. (2020) found small differences between in situ measurements and cryogenically extracted isotope values under natural abundance conditions."

Technical comments:
- L34: "is" instead of "are"

we changed are to "is"

- L61-62: the sentence reads a bit strange, and is somewhat redundant with previous statement. I would suggest to remove it and modify the subsequent one as follows: "Such limitations can be overcome with high-frequency in situ measuring methods [...] community."

We followed your suggestion. Thank you.

- L294: Replace "Tab." by "Table". Same thing on L298

We replaced "Tab." with Table

- L297: "[...] 3 h and 15 h (Quercus), respectively."

we added "respectively"

- L444: "[...] which is especially visible [...]"

We changed the order of the words. Thank you!

- L493-494: Maybe rephrase "[...] differences among the destructive methods we used." ?

Thank you for your suggestion. We rephrased the sentence accordingly: "… we did not find differences among the destructive methods we used."

- L495: since "no trend of differences" is mentioned just before (among destructive sampling), I would suggest to start here with "As compared to in situ measurements, d18O values provided by destructive sampling suggest that the latter contains more [...]"

We followed your suggestion: "As compared to in situ measurements, $\delta^{18}O$ values provided by destructive sampling suggest that the latter contains more tightly bound water that was similar to pre experiment precipitation (Table 2, Fig. 6)."

- L498: Kübert al. (2020) is missing from the bibliography

The reference can be found in the bibliography in L764 (old manuscript)

- L528: Marshall al. (2020) is missing from the bibliography

The reference can be found in the bibliography in L777 (old manuscript)

References
- Kübert, A., Paulus, S., Dahlmann, A., Werner, C., Rothfuss, Y., Orlowski, N., & Dubbert, M. (2020). Water stable isotopes in ecohydrological field research: comparison between In situ and destructive monitoring methods to determine soil water isotopic signatures. Frontiers in plant science, 11, 387.
- Marshall, J. D., Cuntz, M., Beyer, M., Dubbert, M., & Kuehnhammer, K. (2020). Borehole equilibration: testing a new method to monitor the isotopic composition of tree xylem water in situ. Frontiers in plant science, 11, 358.